# Combined Minimally Invasive Treatment of Pyeloureteral Junction Obstruction and Primary Obstructive Megaureter in Children: Case Report and Literature Review

**DOI:** 10.3390/children11040407

**Published:** 2024-03-29

**Authors:** Donatella Di Fabrizio, Irene Tavolario, Lorenzo Rossi, Fabiano Nino, Edoardo Bindi, Giovanni Cobellis

**Affiliations:** 1Pediatric Surgery Unit, Salesi Children’s Hospital, 60123 Ancona, Italy; irene.tavolario@ospedaliriuniti.marche.it (I.T.); lorenzo.rossi@ospedaliriuniti.marche.it (L.R.); fabiano.nino@ospedaliriuniti.marche.it (F.N.); edoardo.bindi@ospedaliriuniti.marche.it (E.B.); giovanni.cobellis@ospedaliriuniti.marche.it (G.C.); 2Department of Specialized Clinical and Odontostomatological Sciences, University Politecnica of Marche, 60121 Ancona, Italy

**Keywords:** pyeloureteral junction obstruction, ureterovesical junction obstruction, high-pressure balloon dilation, pyeloplasty, case report

## Abstract

Introduction: In children, the association of ipsilateral pyeloureteral junction obstruction (PUJO) and ureterovesical junction obstruction (UVJO) is a rare malformation with a non-standardized treatment. We report a case of PUJO and UVJO treated by a combined minimally invasive surgical treatment to resolve the double urinary obstruction. The current literature was also reviewed. Case report and review: A two-month-old boy, without antenatal and postnatal signs of urinary tract anomalies, was hospitalized presenting right hydronephrosis, perirenal fluid effusion, and ascites. An acute pelvic rupture was suspected, and a retrograde pyelogram was performed, showing a primary obstructive megaureter (POM) associated with a corkscrew pyeloureteral junction. The impossibility to place a double J catheter through the pyeloureteral junction led us to achieve percutaneous nephrostomy and an abdominal drain placement. Three months later, the patient underwent a combined high-pressure balloon ureterovesical junction dilation and retroperitoneoscopic Anderson Hynes one-trocar-assisted pyeloplasty (OTAP). The literature search identified 110 children experiencing double urinary tract obstruction. All authors agreed on the difficulty to diagnose both obstructions preoperatively, but there is still no consensus on which obstruction should be relieved earlier, because the alteration in urinary vascularity during a double surgery could damage the ureter. Conclusions: The simultaneous occurrence of UPJO and UVJO is rare, with a challenging diagnosis. Prompt identification and timely surgical intervention are crucial to mitigate the risk of renal function loss attributable to obstruction and infection. Drawing from our expertise and the analysis of the existing literature, we propose employing a simultaneous double minimally invasive strategy in order to optimize the preservation of ureteral vascularity. This approach entails performing a minimally invasive pyeloplasty for the PUJ and utilizing high-pressure balloon dilatation for the UVJ.

## 1. Introduction

Pyeloureteral junction obstruction (PUJO) and ureterovesical junction obstruction (UVJO) are two common causes of hydronephrosis and hydroureteronephrosis in children, and it is well known that they can resolve spontaneously but also can deteriorate. The presence of obstructions at both the proximal and distal ends of the ureter is uncommon, and it must be regarded as a significant condition due to its harmful impact on renal function [1].

The prevalence of combined urinary obstruction documented in the literature ranges from 3% to 25%, and the age of presentation varied from birth to scholar age [2].

The etiopathogenesis of both occurring together is still discussed, and the diagnosis is challenging because the most severe obstruction could mask the second one, revealing the association mostly in the intraoperative or postoperative period [3].

To date, there is no clear agreement on the treatment strategy of this challenging association. Concurrent double junction repair has been discouraged because of the high risk of injury to the vascularity of the remnant ureter. Conversely, a metachronous repair raises the problem of which obstruction should be relieved earlier [4,5].

We report a case of double obstruction presented at our department with an acute rupture of renal pelvis in which a combined high-pressure balloon ureterovesical junction dilatation and an Anderson Hynes retroperitoneoscopic one-trocar-assisted pyeloplasty was successfully performed. We also analyzed the current literature.

## 2. Case Report

A two-month-old boy, without any antenatal and postnatal signs of urinary tract anomalies, was hospitalized for fever and abdominal distension in the pediatric infectious disease department of our hospital. Ultrasound showed a right hydronephrosis (anterior posterior diameter of 1.5 cm), associated with an edematous right kidney with a poor parenchyma differentiation and an important perirenal effusion and ascites; therefore, he was transferred to our department.

Voiding cystourethrography (VCUG) showed neither a vesicoureteral reflux (VUR) nor anomalies on the lower urinary tract. In the case of suspicion of urinary tract rupture, we decided to perform a computed tomography (CT) scan that highlighted an abundant and diffuse ascitic effusion, a right pararenal fluid collection opacified on late urographic scans (Figure 1), a compressed and thread-like renal pelvis, and no ureteric opacification.

In order to treat the acute spontaneous rupture of the renal pelvis and ascites, an abdominal drain was placed through an umbilical incision and a retrograde pyelogram was performed, showing a right primary obstructive megaureter (POM) associated with a corkscrew pyeloureteral junction (PUJ) (Figure 2).

Unfortunately, it was impossible to rise up the JJ ureteral stent to the stenotic PUJ; therefore, a percutaneous nephrostomy was achieved and left in place for one and a half month. Two weeks later, to discern the renal function after the acute hydronephrotic compression and pelvic rupture, he underwent dynamic scintigraphy that showed a symmetric renal function with a right obstructive pattern. To completely understand the anatomy of the right upper urinary tract, a magnetic resonance imaging (MRI) was performed, showing a dysplastic appearance of the PUJ with a coiled junctal ureter and a distal ureter dilatation until the uretero-vesical junction (UVJ) (Figure 3).

Therefore, three months after the acute spontaneous pelvic rupture, the baby boy underwent a combined minimally invasive treatment composed of cystoscopic high-pressure balloon dilatation (HPBD) for the UVJO and one-trocar-assisted pyeloplasty for the PUJO that required a minimal widening of the flank incision up to 2.5 cm. During the postoperative follow-up, there were no signs of urinary tract infection, and there was an improvement in urine drainage, with no evidence of hydroureteronephrosis. The latest ultrasound examination revealed that the right kidney displayed normal corticomedullary differentiation, maintained cortical thickness, and exhibited minimal presence of the renal pelvis, approximately measuring 5–6 mm. Additionally, there was no dilation observed in the kidney’s calyces or ureter (Figure 4).

## 3. Discussion

Congenital anomalies of the kidneys and urinary tracts (CAKUTs) are embryonic conditions that occur during development and lead to a variety of diseases based on the malformation level, such as in the kidneys (e.g., hypoplasia and dysplasia), in the collecting system (e.g., hydronephrosis and megaureter), in the bladder (e.g., ureterocele and vesicoureteral reflux), or in the urethra (e.g., posterior urethral valves) [2]. Above all, the most common form of CAKUT is congenital hydronephrosis, with an estimated incidence of 1/1000 [2]. The first cause of neonatal hydronephrosis is PUJO, followed by POM [3,4], but the association of ipsilateral PUJO and POM in children is very rare, with an incidence ranging from 3% to 25% for all hydronephrosis [5].

Recently, it has been shown that the ureteral maturational process, involving differentiation from epithelial tube to smooth muscle cells and interstitial Cajal’s cells, starts at the mid-ureter during fetal life and extends toward the UPJ and UVJ, and its failure may result in UPJO or UVJO [6]. The basis of this lack of maturation is a delayed molecular mechanism involved in the differentiation of ureteral smooth muscle cells, which could be a T-box transcription factor 18 (Tbx18)-dependent or a Six-1-dependent delay [7,8]. Furthermore, considering the key role of the RAS (renin–angiotensin system) in renal system organogenesis, it has been reported the alteration in AT2R (angiotensin 2 receptor) in American, German, Caucasian, Korean, Italian, and Serbian children with UPJO and POM [8].

Congenital anomalies involving both the ureter junctions (e.g., UPJO and UVJO) frequently lead to abnormal urine transport, such as urinary obstruction (physical or functional obstruction), increasing pressure in the ureter and renal pelvis and eventually leading to their enlargement, which may promote recurrent urinary tract infections, renal scarring, loss of nephrons, and compensatory hypertrophy of remaining nephrons [1].

Conversely, the rupture of the renal pelvis is an exceedingly rare outcome of hydroureteronephrosis, and the literature is very scant [9]. Typically, such ruptures result from trauma or underlying conditions affecting the urinary tract, including tuberculosis, urinary calculi, tumors, or posterior urethral valve abnormalities [10,11,12].

Therefore, even if the diagnosis of double obstruction is not easy, it is important to detect this anomaly in order to prevent the renal function degeneration or pelvic rupture.

Reviewing the literature, we found 110 patients with the association of PUJO and POM reported in 10 articles (Table 1) [3,13,14,15,16,17,18,19,20,21,22].

The available literature showed that the diagnosis of double obstruction was carried out in 67.3% of cases. In the remining 32.7% of patients, the first diagnosis was PUJO in 28 patients and VUJO in 8 patients, and the second obstruction was detected during surgery (36.1%) or during follow-up (63.9%) (Figure 5).

All authors agreed on the difficulty of preoperatively diagnosing both obstructions. The most severe obstruction at one end of the ureter may completely mask the second obstruction: an important obstruction at the PUJ may keep an obstructed ureter from dilating; furthermore, a high-grade obstruction at the UVJ may cause severe hydroureteronephrosis such that the PUJ stenosis is overlooked [13,14]. Indeed, the diagnosis of double obstruction is usually reported intraoperatively during pyeloplasty, when the surgeon faces difficulty in passing double J stent across the UVJ, or postoperatively, because of the appearance of hydroureteronephrosis. Therefore, it has been strongly recommended that, if there is any doubt regarding the presence of double obstruction, a retrograde ureteropyelography, to look for the ureter and its drainage, should be performed [5,19].

In our case, the diagnosis of a double obstruction was achieved intraoperatively, before the pyeloplasty, thanks to the spontaneous pelvic rupture: the hydronephrosis and the pelvic rupture implied a PUJO, but the intraoperative retrograde ureteropyelography showed a POM with a corkscrew PUJ, confirmed by the impossibility to pass the JJ stent.

It is not known whether the two obstructions developed together or if one was the cause for the other [18,21]. Ebadi et al. hypothesized that the ureteral kinking caused by the obstruction at the UVJ level produced a secondary PUJO; therefore, by first approaching UVJO, there is a chance for spontaneous resolution of the concomitant PUJO because it eliminates the primary cause [18].

However, Reich et al. had a patient with bilateral POM, treated with bilateral ureteral reimplantation, who after surgery developed a bilateral PUJO, and it has been postulated that an alteration in the ureteral vascularity during surgery could play a key role in the appearance of a second obstruction [21].

Conversely, other authors argued that since a certain number of POMs can be expected for resolution, pyeloplasty should be the first procedure to perform [13,14,15], and only if the UVJO persists, reimplantation should be carried out because of the risk of ureteral vascularity damage.

The literature review showed that the authors treated a single obstruction in 95.5% of cases, including pyeloplasty in 41.8% of children, ureteral reimplantation in 10.9% of children, and endoureterotomy in 42.7% of children. Among them, 42% of patient later on required a second surgery, including pyeloplasty (26.8%), ureteral reimplantation (51.2%), endoureterotomy (19.5%), and surgery on both obstructed junctions (2.4%).

The combined treatment of both obstructions as first surgery occurred only in 3.6% of patients (Figure 5). These results underline that almost 50% of patients with double obstruction will require operative treatment for both junctions.

There is still no consensus on which obstruction should be relieved earlier, but minimally invasive endourology techniques could represent a good solution in this challenging situation.

Endoscopic interventions have emerged as pivotal methods in treating POM, particularly in infants and children. Procedures such as HPBD and endoureterotomy are carried out via the urinary tract without necessitating skin incisions. These techniques offer a viable option for releasing obstruction and facilitating urinary drainage without the need for excising the distal ureteral segment [22].

Specifically, HPBD of the ureterovesical junction has surfaced as a definitive therapeutic approach for POM in pediatric patients, demonstrating satisfactory success rates and minimal occurrences of complications (mainly transient hematuria, urinary tract infections, and occurrences of stent intolerance or migration) [22]. It has comparable success with classic reimplantation (86% vs. 91%), but it offers a minimally invasive approach that is safe and feasible even for patients under 1 year of age [23,24].

In our case, HPBD allowed the resolution of the UVJO, preserving the ureteral vascularity, and it was possible to perform it at the same time as the pyeloplasty for the PUJO [25].

Additionally, attempts have been made to combine EBD with endoureterotomy or perform it without fluoroscopic guidance. Endoureterotomy has been conducted utilizing either a pure cutting electrical current or laser technology, reaching satisfactory success rates of around 90%, coupled with a minimal occurrence of complications, and presenting as well as HPBD, a minimally invasive substitute to traditional open surgical interventions [22]. In particular, if the stenotic ring is still visible after HPBD, endoureterotomy could be considered, being a technique that allows to keep the vascularity intact and allows a simplified healing of the muscular layer with the JJ stent left in place [18,23].

## 4. Conclusions

The simultaneous occurrence of UPJO and UVJO is rare, and it is evident that diagnosing cases with double urinary obstruction in the same ureter can be challenging. Early diagnosis and timely surgical intervention are essential to prevent the loss of renal function resulting from obstruction and infection. Based on our experience and the literature analysis, we suggest using a simultaneous double minimally invasive approach composed of a minimally invasive pyeloplasty for the PUJ joint and a high-pressure balloon dilatation for the UVJ, in order to maximum the preservation of the ureteral vascularity.

## Figures and Tables

**Figure 1 children-11-00407-f001:**
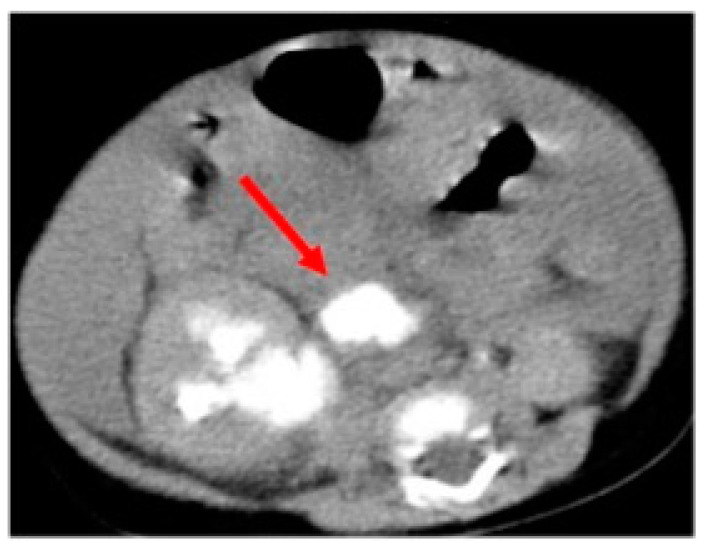
Computed tomography scan underlining the spontaneous pelvic rupture with the pararenal fluid collection opacified on late urographic scans (arrow).

**Figure 2 children-11-00407-f002:**
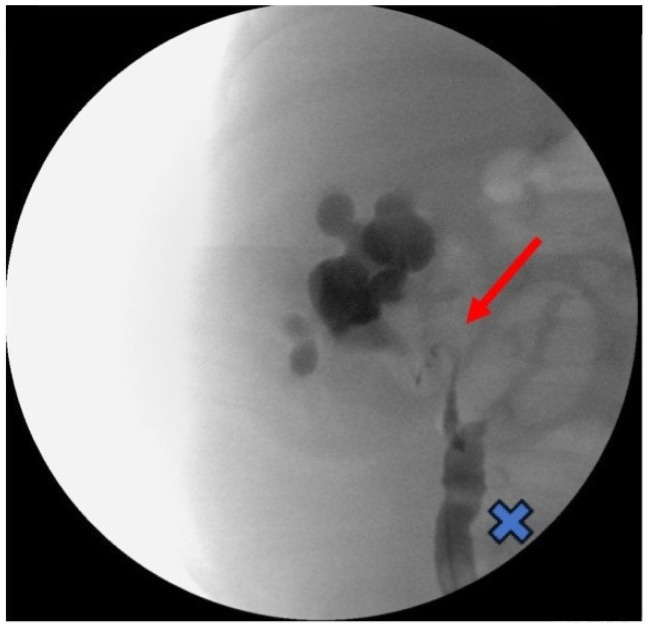
Intraoperative retrograde pyelogram showing a right primary obstructive megaureter (cross) associated with a corkscrew pyeloureteral junction obstruction (arrow).

**Figure 3 children-11-00407-f003:**
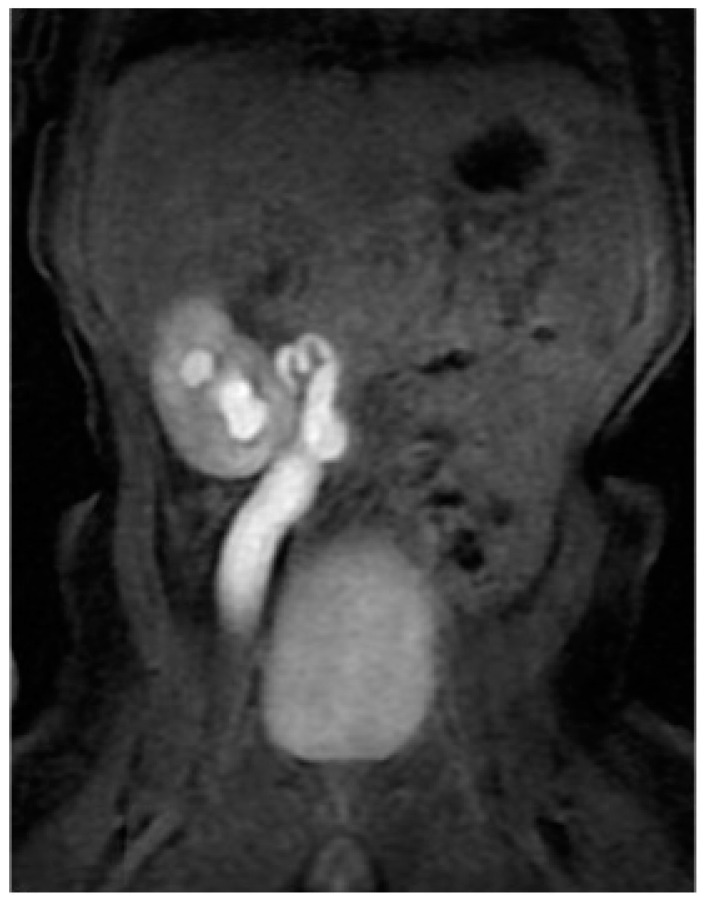
Hydronephrosis on a corkscrew pyeloureteral junction with primary obstructive megaureter at the magnetic resonance imaging.

**Figure 4 children-11-00407-f004:**
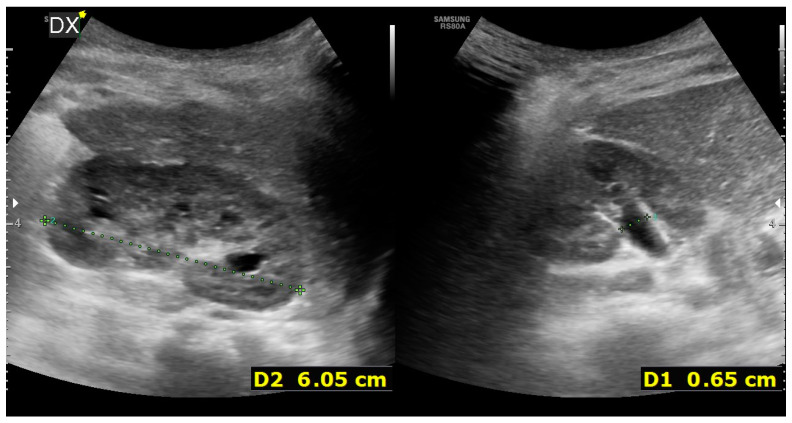
US 8 months after surgery showing the right kidney with regular cortico-medullary differentiation, preserved cortical thickness, renal pelvis of approximately 5–6 mm, and no dilation of the calyces or the ureter.

**Figure 5 children-11-00407-f005:**
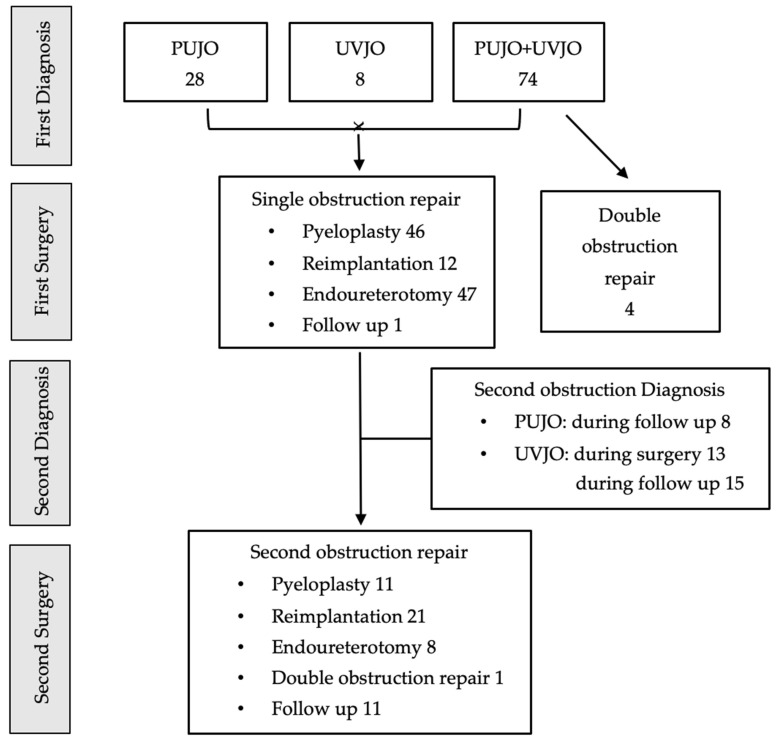
Diagnostic and operative management from literature review.

**Table 1 children-11-00407-t001:** Literature review.

Article			Demographics			First Diagnosis				First Surgery		Second Diagnosis				Second Surgery	
Author	Years	Country	Patients N°	Sex (M: Male, F: Female)	Age	Side (L: Left, R: Right, B: Bilateral)	PUJO	UVJO	PUJO + UVJO	Procedure	Others	PUJO		UVJO		Procedure	Others
												During Surgery	During Follow-Up	During Surgery	During Follow-Up		
McGrath	1987	USA	14	7 M, 7 F	0–11 years	11L, 3R	5	6	3	Pyeloplasty (5), ureteral reimplantation (6), and simultaneous operation (2)	Follow-up for both obstructions (1)		6		5	Reimplantation (2) and pyeloplasty (6)	Follow-up (3)
Pesce	2003	Italy	11	7M, 4 F	0–8 years	n/a	6	/	5	11 pyeloplasty		/	/	3	3	Reimplantation (6)	
Cay	2006	Turkey	14	10 M, 4 F	2–54 months	9L, 5R	9	/	5	14 pyeloplasty		/	/	6	3	Reimplantation (4)	Follow-up (1)
Moodley	2010	Canada	2	1 M, 1 F	0–6 weeks	1L, 1R	2	/	/	2 pyeloplasty	Cold knife ureterocele incision (2)	/	/	2			
Neulander	2010	Israel	1	1 M	20 days	1L	/	/	1	Percutaneous drainage and later on upper pole partial nephrectomy and pyeloplasty with a modified Y urethrotomy		/	/	/	/	Reimplantation (1)	
Ebadi	2013	Iran	47	19 M 28 F	n/a	27R, 20L	/	/	47	Endoureterotomy (47) (VUJO)	39/47 successful for both obstructions	/	/	/	/	Second attempt of endoureterotomy (8)	Resolution of both obstructions (3), ureteral reimplantation (2), reimplantation + mini-pyeloplasty (1), and mini-pyeloplasty (2)
Lee	2014	Korea	15	n/a	n/a	n/a	4	1	10	Pyeloplasty (9), ureteroneocystostomy with Starr plications (5), both procedures (1)			1		4	Ureteroneocystostomy (2) and pyeloplasty (2)	
Sharma	2014	India	4	n/a	n/a	n/a	2	/	2	Pyeloplasty (4)		/	/	2		Reimplantation (4)	
Kumar	2020	India	1	F	16 years	L	/	/	1	Robotic pyeloplasty and reimplantation		/	/	/	/		
Reich	2023	USA	1	F	11 months	B	/	1	/	Bilateral Politano-leadbetter ureteral reimplantation	Nephrostomy left in place		1	/	/	L pyeloplasty, R pyeloplasty (1 month later)	

## Data Availability

Not applicable.

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
