# Peer review of "Combined Minimally Invasive Treatment of Pyeloureteral Junction Obstruction and Primary Obstructive Megaureter in Children: Case Report and Literature Review"

_children, 2024, doi:10.3390/children11040407_

Round 1

Reviewer 1 Report

Comments and Suggestions for Authors

The main question addressed by the research is how to better diagnose and manage rare entity of double obstruction. Originality is in the unique case of double obstruction, not infravesical (what I asked to cinfirm with more data) and combined with spontaneous rupture. This work add Unique rarity and contemporary management. One troacar assisted pyloplasty is not my choice but in the conclusions authors only say miniinvasive, which is correct. More date in case description and follow-up should be improved. For the case report with literature review, conclusions are formulated correctly. References are appropriate.

Very interesting case but I would ask for more discussion and literature review as to the occurrence of spontaneous rupture. It is usually reported in the context of infravesical obstruction or trauma. In the case description urinary tract infection or bladder function information is missing. It is stated that VCUG was normal but bladder on MRI image looks "hyperactive", although difficult to judge. How long was follow-up? Minor corrections in the text:

Primary, not primitive, obstructive ureter in the abstract

Diagnosis, not diagnose, repeatedly

Author Response

Thank you for your kind observations.

We improved data about follow up and we added the last US.

There were no urinary infections after surgery, and the bladder function was not explored because he had no problems in urinary output, the VCUG was normal and the renal function at the dynamic scintigraphy was symmetric.

Furthermore, we implemented the discussion.

Reviewer 2 Report

Comments and Suggestions for Authors

Evaluation report.

Thank you for sending me this manuscript for consideration. This study is well prepared and presented. I think that this study will make an important contribution to the literature.  However, some revisions should be made. We encountered a similar patient in our clinic. We performed a total of two anastomoses on a patient who had double obstruction (UPJ and UVJ). During the patient's follow-up, he developed a urinoma. While performing surgery, we noticed that the ureteral tip on the lower side had been necrotized. As the ureter was too short to reach the bladder, We anastomosed the short ureter to the other ureter with a crossover. We did not publish this case. I understand what a difficult situation authors face.

1-This work has been good, but there are important deficiencies. Is there an examination, radiological image or USG results that prove that the patient has recovered after the treatment? Such as kidney size, ureter diameter, parenchymal thickness, renal pelvis diameter.

2-How did they understand that the source of the ascites present in such a patient was due to a spoantan rupture? If it were me, I would look at the results of BUN and creatin in the blood biochemistry test. Because of the absorbed urine in the peritoneum, they increase. The results of post-rupture Bun creatin may be important to describe this patient. More importantly, a good message is given to the reader.

3-The most important part of this study is to teach how to follow an algorithm in spontaneous urinary rupture. See, there is a publication about calyceal rupture in this way. Let's add this to the discussion. (Hunley TE, Adams MC, Hernanz-Schulman M, Jabs K. A critically ill newborn with a distended abdomen. Kidney Int. 2017 Aug; 92(2):521. doi: 10.1016/j.kint.2017.03.001. PMID: 28709610.) In this publication, the values of BUN and creatin were found to be significantly higher. Please add the values of the BUN creatin and indicate that it is elevated due to absorption from the peritoneum.

Comments on the Quality of English Language

Evaluation report.

Thank you for sending me this manuscript for consideration. This study is well prepared and presented. I think that this study will make an important contribution to the literature.  However, some revisions should be made. We encountered a similar patient in our clinic. We performed a total of two anastomoses on a patient who had double obstruction (UPJ and UVJ). During the patient's follow-up, he developed a urinoma. While performing surgery, we noticed that the ureteral tip on the lower side had been necrotized. As the ureter was too short to reach the bladder, We anastomosed the short ureter to the other ureter with a crossover. We did not publish this case. I understand what a difficult situation authors face.

1-This work has been good, but there are important deficiencies. Is there an examination, radiological image or USG results that prove that the patient has recovered after the treatment? Such as kidney size, ureter diameter, parenchymal thickness, renal pelvis diameter.

2-How did they understand that the source of the ascites present in such a patient was due to a spoantan rupture? If it were me, I would look at the results of BUN and creatin in the blood biochemistry test. Because of the absorbed urine in the peritoneum, they increase. The results of post-rupture Bun creatin may be important to describe this patient. More importantly, a good message is given to the reader.

3-The most important part of this study is to teach how to follow an algorithm in spontaneous urinary rupture. See, there is a publication about calyceal rupture in this way. Let's add this to the discussion. (Hunley TE, Adams MC, Hernanz-Schulman M, Jabs K. A critically ill newborn with a distended abdomen. Kidney Int. 2017 Aug; 92(2):521. doi: 10.1016/j.kint.2017.03.001. PMID: 28709610.) In this publication, the values of BUN and creatin were found to be significantly higher. Please add the values of the BUN creatin and indicate that it is elevated due to absorption from the peritoneum.

Author Response

  1. Thank you for your comment. We added the US done 6 months after surgery showing the right kidney with regular cortico-medullary differentiation, preserved cortical thickness, minimal evidence of the renal pelvis of approximately 5-6 mm and no dilation of the calyces or ureter.
  2. Thank you for your kind observation.

    We checked BUN, Creatinine and glomerular function during the whole hospital stay, but the diagnosis of rupture was performed mainly on imaging, because at the very beginning they were normal but when we had the urinary rupture, we drained it the day after the rupture and on the blood exams we had BUN and Creatinine that were low since the nephrostomy while the glomerular function stayed normal during the whole hospitalization.

  3. Thank you so much for your suggestion. We added the article to the discussion but, the aim of the paper was to report a case of double obstruction, so we focused the discussion about the description of all we know in literature about the combined obstruction of urinary tract. The description of a diagnostic and therapeutic  algorithm in spontaneous urinary rupture is not the objective of the study, but it could be and interesting suggestion for another article.

Round 2

Reviewer 2 Report

Comments and Suggestions for Authors

The authors fulfilled their responsibilities. This work can be published.